# Multiscale Simulation of Branched Nanofillers on Young’s Modulus of Polymer Nanocomposites

**DOI:** 10.3390/polym10121368

**Published:** 2018-12-10

**Authors:** Shengwei Deng

**Affiliations:** College of Chemical Engineering, State Key Laboratory Breeding Base of Green-Chemical Synthesis Technology, Zhejiang University of Technology, Hangzhou 310014, China; swdeng@zjut.edu.cn; Tel.: +86-155-5713-0167

**Keywords:** reinforcement, dispersion, lattice spring model, stress distribution

## Abstract

Nanoscale tailoring the filler morphology in experiment offers new opportunities to modulate the mechanical properties of polymer nanocomposites. Based on the conventical rod and experimentally available tetrapod filler, I compare the nanofiller dispersion and elastic moduli of these two kinds of nanocomposites via molecular dynamics simulation and a lattice spring model. The results show that the tetrapod has better dispersion than the rod, which is facilitate forming the percolation network and thus benefitting the mechanical reinforcement. The elastic modulus of tetrapod filled nanocomposites is much higher than those filled with rod, and the modulus disparity strongly depends on the aspect ratio of fillers and particle-polymer interaction, which agrees well with experimental results. From the stress distribution analysis on single particles, it is concluded that the mechanical disparity between bare rod and tetrapod filled composites is due to the effective stress transfer in the polymer/tetrapod composites.

## 1. Introduction

The incorporation of rigid particles with dimensions in the nanometer or micrometer range into polymers affords engineers an opportunity to design polymer composites with optimized mechanical properties [1,2]. Many studies [3,4] have been conducted in this field because of its technological and scientific importance. Undoubtedly, the mechanical properties of particulate-polymer composites are affected by the particle size [5,6], shape, loading and particle-matrix interface adhesion [7,8]. With the availability of controlled manipulation to obtain uniform-sized fine structures [9,10], the filler morphology is becoming increasingly important to modulate the mechanical behavior.

Different filler morphologies usually result in the disparity of mechanical properties in corresponding polymer composites [11,12]. For example, platelets are more efficient for reinforcing composites than rods and spheres under the same conditions [13]. As a shape-isotropic filler, the tetrapod particle [14,15] is usually made of various semiconductor materials and shows significant potential in the electrical [16] and optical [17,18] applications. The tetrapod has four pods binding together in one point as the center and stretching to each corner of the tetrahedron [19]. This peculiar shape may potentially be an alternative to fibers or rods as additives for mechanical reinforcement of polymers [20,21]. Recently, Alivisatos et al. prepared the nanoscale tetrapod with surface treatment and showed its superior ability in mechanical reinforcement of polymer nanocomposites as fillers, which was mainly contributed by the orientation of the strong X-type bonds at the nanoparticle-ligand interface [22]. To avoid the influence of alkyl chain ligands and examine the shape effect solely, another work claims that tetrapod as filler is more efficient to enhance the elastic modulus of polymer than rod as filler [23], which is probably due to the difficulty of reorientation and low packing density of tetrapod. Furthermore, the tetrapod varieties, such as three-dimensional aerographite networks [24] which are built from interconnected hollow tubular tetrapods, show their challenging mechanical properties [25], e.g., extremely robust to bear strong deformations [26]. Inspired by these works, one would think about the intrinsic factor of the superiority of tetrapod filler in the mechanical enhancement, which receives only limited attention.

To show the advantage of tetrapod filler on the mechanical reinforcement, it is reasonable to compare the mechanical behavior of tetrapod filled polymer composites with the other branched particle filled ones. Rod-shaped filler is one of the most common fillers to offer the possibility of substantial mechanical improvements [27,28], which is suitable for this comparison. The experimental study has been playing a pivotal role in this field since the invention of polymer nanocomposites. However, the understanding the shape effect at nanoscale is largely empirical, and a finer degree control of the structure-property relationship is not easy to achieve so far. On the other hand, the accuracy of computer simulation for mechanical behaviors has improved greatly to obtain a reasonable interpretation of experimental results. In particular, the simulation can precisely examine the effect of a single factor (branch type) on the final mechanical performance, accompanied with the analysis of local microstructure evolution [29,30].

In order to efficiently describe the elastic deformation and local stress/strain distribution, numerical solutions for analysing the continuum mechanics of a material’s microstructure are required. For the purpose here, it is time-consuming and unnecessary to perform large-scale computer simulations at the atomic or molecular scale, though these small-scale methods could well predict some macroscopic properties of solids [31]. The lattice spring model (LSM) is adopted from condensed matter physics as a highly efficient method of discretizing continuum media, and is shown to be algebraically equivalent to a simple finite element method (FEM) [32]. This method is particularly well suited to investigating the physics of stress transfer and stress/strain distribution in complicated heterogeneous systems (such as polymer nanocomposites). The simulation of heterogeneous materials by LSM is achieved through assigning force constants to individual bonds depending upon the composition on each grid and thereby avoiding complicated mesh generation in FEM. However, the Born LSM adopted in this work [32,33] is only valid under small strain, and the input structure is considered as the equilibrium structure, where the LSM cannot be used to study the micro-structural evolution. Therefore, micro-scale or meso-scale simulation methods (such as molecular dynamics (MD) simulation [34]) are required to study the spatial organization of particles. To clarify the prominent enhancement effect of tetrapod filler on the final mechanical properties, an innovative strategy is adopted to reveal the intrinsic mechanism, which involves the local stress analysis by highly efficient 3D LSM and spatial distribution prediction by MD simulation. I hope that this effective prediction strategy could apply to more types of fillers (such as interconnected fillers) and provide theoretical backgrounds for rational design of polymer nanocomposites.

In the present work, based on tetrapod or rod particle as filler in the composites, the mechanical response is modelled by continuum mechanics based LSM, while the dispersion state is simulated by a coarse-grained MD method. The stress transfer between particle and polymer matrix is studied in detail with the variation of particle size, shape, intrinsic material properties and particle-polymer interaction (NPI). Meanwhile, the dispersion and spatial organization of particles in polymer melts are examined with various NPIs.

## 2. Model and Simulation Methods

### 2.1. Lattice Spring Model (LSM)

Born LSM [35,36] is a numerical technique for discretizing the linear elasticity theory. Here, a material is represented by a network of springs that occupy the nearest and next nearest neighbour bonds of a simple cubic lattice. Although the LSM model lacks rotational invariance, it is useful in the study of heterogeneous systems because the next nearest neighbour’s central and bond-bending interactions can be locally changed to account for local variations in the stiffness of heterogeneous systems. Typically, the LSM allows one to determine micromechanical properties in a computationally efficient way.

The energy associated with a node *m* in the cubic lattice is given by(1)$$E_{m}=\frac{1}{2}\sum_{n}(u_{m}-u_{n}{)}^{T}\cdot M_{mn}\cdot (u_{m}-u_{n}),$$
where *n* is the neighbouring nodes connected to *m* by a spring, the vector um is the displacement of the mth node from its original position, Mnm is a symmetric 3 × 3 tensor which describes the interaction between various nodes through central and noncentral force constants. For example, the matrix associated with the spring in the [101] directions are of the form(2)$$M_{\left[101\right]}=\frac{1}{2}(\begin{array}{ccc} k+c & 0 & k-c\\ 0 & 2c & 0\\ k-c & 0 & k+c \end{array}).$$

The matrix for the remaining 17 directions can be obtained by a similarity transformation [32]. The central force constant energetically compensates for the spring extension, while the noncentral force constant penalizes the rotation of springs from their original orientation. The central force constant *k* and noncentral force constant *c* take the following forms:(3)$$k=\frac{E}{5(1-2\nu )},\;c=\frac{E(1-4\nu )}{5(1+\nu )(1-2\nu )},$$
where *E* and v are the Young’s modulus and Poisson’s ratio, respectively.

The elastic force acting on the mth node is a linear function of the displacement because of the harmonic form of energy in Equation (1). The force acting on the mth node, due to the local displacement of the spring between nodes *m* and all neighbouring *n*, is given by(4)$$F_{m}=\sum_{n}M_{nm}\cdot (u_{m}-u_{n}).$$

If the external forces are applied to the boundary nodes with the spring constants specified, the constraint that all these linear forces must balance at each node at equilibrium results in a set of sparse linear equations. The solution of these equations is obtained by using a conjugate gradient method to find the equilibrium configuration corresponding to the situation without a net force at each node. The stress and strain tensors are calculated using the forces and displacements. The strain tensor is obtained by using a finite difference approximation of the displacement field. A central difference approximation is given by(5)$$\delta_{x}u_{\left(i,j,k\right)}=\frac{\left[-u_{\left(i+2,j,k\right)}+8u_{\left(i+1,j,k\right)}-8u_{\left(i-1,j,k\right)}+u_{\left(i-2,j,k\right)}\right]}{12h},$$
where ***u***_(*i,j,k*)_ is the displacement field at coordinates *i*, *j* and *k*. The *x* represents the tensile direction. The *ij*th component *σ_m,ij_* of the stress tensor acting on the central node *m* of a cubic unit cell is defined as(6)$$\sigma_{m,ij}=\frac{\sum_{s}F_{m,s}\cdot p_{ij}^{m,s}}{A},$$
where ***F****_m,s_* is the force acting on a surface *s* across node *m* of the cubic cell, *A* is the area of surface perpendicular to the spring of tensile direction in the cubic cell, ∑_s_ represents a sum over all the cube surfaces across node *m*, while ***p*** is a unit vector with its components either normal or parallel to the surface *s.* The average strain and the applied stress can then be used to calculate the Young’s modulus, which is defined as the stress of a material divided by its strain. For more details, see Ref. [37,38,39,40].

#### Spring Network in Particle Filled Composites

Based on the composites structure filled with rod with aspect ratio of 5, Figure 1 shows a layer of interconnectivity across the branched particle. Two kinds of beads are shown in different colors. The spring constant depends on the bead on both sides of spring. Here, three kinds of spring constants are considered in the model: polymer and polymer (*E*_pp_), polymer and particle (*E*_np_) and particle and particle (*E*_nn_). In the surface modified particle filled system, the surficial ligand exists between polymer and particle, and two new spring constants need to be defined.

In the initial structure, the particle is randomly dispersed in the polymer matrix under periodic boundary conditions. During the mechanical test, a uniaxial strain field equivalent to 1% global strain is applied and the system equilibrated. Each elastic modulus value in the following figures takes an average of 10 tests.

### 2.2. Molecular Dynamics Simulation

All simulations were performed with the help of parallelized LAMMPS code [41]. I employed a coarse-grained bead spring model [34] to capture the key physics of polymer nanocomposites. The idealized polymer consists of thirty beads. The total number of simulated polymer beads is 8100. Each nanorod consists of five beads, and each tetrapod contains 17 beads. The total numbers of rod are 34, 68, 136, 204, 272, 344, 442 or 544; accordingly, the total numbers of tetrapod are 10, 20, 40, 60, 80, 100, 130 or 160. These short chains here are helpful for the improvement of computational efficiency and can also capture the static and dynamic behavior characteristics of long chains [42]. All non-bonded beads interact through a truncated Lennard-Jones potential:(7)$$\begin{array}{ll} U_{IJ}\left(r\right)=0, & r\geq r_{\mathrm{cutoff}}\\ U_{IJ}\left(r\right)=4\varepsilon \left[{\left(\sigma /r\right)}^{12}-{\left(\sigma /r\right)}^{6}\right]+C, & r<r_{\mathrm{cutoff}} \end{array},$$
where *C* is a constant that satisfies that the potential is continuous everywhere (*C* = 4*ε*_np_[(*σ*/*r*_cutoff_)^6^ − (*σ*/*r*_cutoff_)^12^]). In the following, *ε* = 1 and *σ* = 1 are taken as units of energy and length, respectively. Note that all calculated quantities are dimensionless. *r*_cutoff_ stands for the distance at which the interaction is truncated and shifted. The polymer-polymer interaction parameter and its cutoff distance are *ε*_pp_ = 1.0 and *r*_cutoff_ = 2.24 and the particle-particle interaction parameter and its cutoff distance are *ε*_nn_ = 1.0 and *r*_cutoff_ = 1.12, while the polymer–nanorod interaction parameter *ε*_np_= 0.5, 1.5 or 4.0 with its cutoff distance *r*_cutoff_ = 2.5. These force filed parameters (such as *ε*_np_, *ε*_nn_, *ε*_pp_ and *r*_cutoff_) are adopted from Refs. [42,43]. Figure 2 shows the plot of interaction energy versus distance with different interactions, where the lowest points of energy are all located at the distance of 2^1/6^ ≈ 1.12. The pure repulsive interaction between nanoparticles is applied to avoid the particle aggregation. For the polymer-particle interaction, the higher *ε*_np_ leads to stronger interaction.

Covalent bonds between adjacent beads for both polymer chains and particles are modeled using the finitely extensible nonlinear elastic (FENE) potential:(8)$$U_{FENE}\left(r\right)=-0.5KR_{0}^{2}\ln\overset{}{\left[1-{\left(r/R_{0}\right)}^{2}\right]},$$
where *R*_0_ = 1.5*σ* and *K* = 30*ε*/*σ*^2^. The chain is unbreakable in this work.

To maintain the structure of rod or tetrapod, a bending potential is applied to particles, given by(9)$$U_{\mathrm{angle}}=K_{a}\overset{}{\left[1-\cos\left(\theta -\theta_{0}\right)\right]},$$
where *θ* is the bending angle formed by three consecutive rod beads, and *K_a_* is equal to 300. For the angle in the nanorod and straight arm in tetrapod, *θ*_0_ = 180°, and, for the angle between different arms in tetrapod, *θ*_0_ = 109.5°. The equations of motion are integrated with the velocity-Verlet algorithm and a time step d*t* = 0.01*t*_LJ_ (*t_LJ_* = (*mσ*^2^/*ε*)^1/2^, *m* = 1 is the bead mass). Temperature is controlled with a Nosé-Hoover thermostat.

The initial structure was built by self-avoiding random walk method, and periodic boundary condition was imposed in all directions. NPT (Isothermal–isobaric and constant number of particles) ensemble is adopted until the number density of polymers becomes greater than 0.8, where the temperature and pressure were fixed at *T* = 1.0 and *P* = 2.0, respectively. Further equilibration under *NPT* with *T* = 1.0 and *P* = 0 was performed. The equilibration temperature here is well above the glass transition temperature for bulk polymers *T*_g_ ≈ 0.56ε/*k*_B_ [42]. More detailed simulation processes can be found in Ref. [43].

## 3. Results

The mechanical response of polymer composites is modeled by 3D LSM, while the equilibrium structure is simulated by molecular dynamics method.

### 3.1. Nanocomposites Filled with Branched Particles with Surface Ligands (BPSL)

The model systems in LSM are based on (CdSe/CdS)/SEBS nanocomposites [44] (SEBS: poly(styrene block-ethylene-butylene block-styrene)), and the fillers in these composites are nanoscale nanorods or tetrapod quantum dots with surface ligands. The bond between ligands and nanoparticles is a strong X-type bond, which is a strong interaction of an anionic phosphonate/phosphinate moiety with a surface Cd^2+^ ion [45]. To fit for the real experiment, the spring constant between ligand and nanoparticle (*E*_nl_) is 1000 times greater than *E*_pp_. According to the literature value [22], *E*_nn_ = 2000*E*_pp_. The ligand/polymer spring constant is critical to the overall mechanical reinforcement. Based on the assumption in 2D LSM [22], which gives the best agreement between theory and experiment, I used the same spring constant relationships in our 3D model, *E*_lp_ = 0.75*E*_pp_ and *E*_ll_ = 0.5*E*_pp_. Previous results show that the tetrapod with surface ligands has superior ability in mechanical reinforcement than the rod as fillers [22].

Inspired by Guth’s equation [46], which claims that the modulus of rod-like particle reinforced elastomers is based on the aspect ratio and volume fraction of the filler, I measured the elastic modulus for rod and tetrapod nanocomposites with the increase of filler’s aspect ratio. I decrease the total number of particles to ensure that the number of X-type bond is almost unchanged with the increase of the aspect ratio. Figure 3 shows that the relative elastic moduli for both rod and tetrapod nanocomposites increase with the increase of aspect ratio. Especially for the tetrapod nanocomposites, the elastic modulus is influenced by the aspect ratio. It indicates that the mechanical disparity between branched rod and tetrapod as fillers is also related to the filler shape.

### 3.2. Nanocomposites Filled with Barely Branched Particles

To study the shape effect solely, I prepared the composites filled with bare rod or tetrapod based on ZnO/PDMS composites [23] (PDMS: Polydimethylsiloxane). According to the literature value for the Young’s modulus of ZnO [47] and PDMS [23], *E*_nn_ = 25000*E*_pp_. To give the best agreement between simulation and experiment, *E*_np_ = 17.5*E*_pp_ is chosen. Figure 4 shows that the elastic modules of composites increase with the increase of particle concentration. The simulation data fits quite well with the experimental data. The tetrapod filler is more efficient to enhance the elastic modulus of composites than the rod at the same particle concentration. All particles here are bare particles without surface ligands, which indicates that solely the shape difference can also lead to different mechanical enhancement effects. Therefore, the shape effect might be considered to explain the mechanical disparity between rod and tetrapod filled composites.

### 3.3. Effect of Ssurface Treatment of Particle on the Mechanical Reinforcement

Based on the analysis above, the shape difference is a factor to lead to the mechanical disparity, while the effect of surface treatment (ligands) is still unclear. Using the same input parameters in LSM as defined in Section 3.1 (Figure 3), I compared the elastic modulus of composites filled with bare particles or BPSL. The interaction between polymer and particle in bare particle composites is the same as the interaction between ligand and polymer in BPSL composites. The blue line in Figure 5 shows the relative elastic modulus of BPSL filled composites, while the black line shows the result from bare particle filled composites. Apparently, the BPSL is much more efficient to reinforce the composites comparing to the bare particle. It means that the surface ligands on the particle leads to the amplification of this disparity. One can also claim that this disparity for nanocomposites filled with BPSL is mainly contributed by the orientation of the strong bonds at the nanoparticle−ligand interface [22]. From the micromechanical point of view, surface treatment is also a way to enhance the interaction between particle and polymer matrix. It may correspond to the strong restriction of polymer chains near the branched particle in polymer composites.

### 3.4. Stress Distribution on a Single Bare Particle and Intrinsic Mechanism of Filler Effect

The shape difference will lead to the mechanical disparity for bare particle filled composites. To reveal the intrinsic mechanism based on this phenomenon, I plot the local stress distribution on single particle by using LSM. As shown in Figure 6, the left three figures (a, c, e) show the stress distribution of bare particle filled composites with weak particle-polymer interaction (NPI), and the right three figures (b, d, f) show the stress distribution of particle filled composites with strong NPI. The fourth arm in tetrapod cannot be shown in this two-dimensional profile, and the color bar on the right shows the relative value of the stress. After deforming the rod filled sample at the same strain, the stress is not concentrated on the vertically aligned rod (Figure 6c), while the stress concertation is observed on a horizontally aligned rod (Figure 6e). Hence, the horizontally aligned rod is more efficient to reinforce the composite than a vertically aligned rod. The effective elastic modulus for rod composites is a combination of elastic modulus of vertically aligned rod composites and horizontally aligned rod composites. Due to the isotropic structure of tetrapod, it is impossible to let all arms align perpendicular to the tensile direction. Figure 6a shows the stress concertation on each arm in tetrapod, even though the arm is not horizontally aligned along the tensile direction. Because of the strong connection between each arm in a tetrapod, the stress will transfer from the strong arm (relatively horizontally aligned arm) to a weak arm (relatively vertically aligned arm); this results in the overall enhancement of the stress on the filler. This stress transfer process finally leads to the mechanical disparity between rod and tetrapod composites. Comparing the stress concertation on particles with strong or weak NPI, the value of the stress is significantly enhanced with the increase of NPI. It means that composites with strong NPI are much more efficient to improve the elastic modulus of corresponding composites and amplify the disparity between rod and tetrapod composites [48].

It is clear that the final Young’s modulus of polymer composites is strongly affected by many factors including the filler shape, aspect ratio, NPI and intrinsic material properties. Figure 7 shows the enhancement of Young’s modulus with the variation of different impact factors. These factors are permeating and influencing one another, and thus, for the rational design of nanocomposites, it is difficult to modulate a factor without variation of other factors. Notably, the strong NPI and uniform distribution seem like the basis for mechanical improvement unless the fillers are interconnected and filled with the whole matrix.

### 3.5. Dispersion State Comparison of Rod with Tetrapod as Fillers

In the simulation by LSM, the initial structures are generated by the random dispersion method. The particle agglomeration is not considered in the above analysis. However, the mechanical properties of composites are strongly relying on the dispersion state of fillers. Nanofillers agglomeration usually negatively affected mechanical behavior of the resulting composites, e.g., particle aggregation in polymer composites leads to only a slight increase or even a decrease [56] of the elastic modulus compared to the pristine polymer. Here, coarse-grained molecular dynamics is adopted to study the dispersion and aggregation mechanisms of rods and tetrapods in polymer melts. When the polymer-filler interaction (*ε*_np_) is lower than the filler–filler interaction (*ε*_nn_ = 0.5), the direct contact aggregation of particles occurs in both rod and tetrapod composites. To further characterize the nanoparticle dispersion state, I plot the inter-nanoparticle radial distribution function (RDF) and RDF between nanoparticle and polymer beads in Figure 8. The results are based on systems with nanoparticle concentration of 17.3%. The first peak of “weak” (*ε*_np_ = 0.5) interaction systems in Figure 8a is around 1.12, which corresponds to the distance of nearest neighbors. It confirms that the nanoparticle is aggregated at *ε*_np_ = 0.5. In “moderate” (*ε*_np_ = 1.5) and “strong” (*ε*_np_ = 4) interaction systems, the first peak is around 2, which corresponds to the second neighboring beads; this means that the direct contact of nanoparticle does not occur under these two interactions. In addition, the first peak of tetrapod nanocomposites is much lower than that of rod nanocomposites; this can only be explained by the loose aggregation of tetrapod systems. Note that the local bridging [57] via polymer beads in rod systems can be observed with further increase of the polymer-particle interaction. From the neighboring polymer beads distribution in Figure 8b, the first peak (around 1.12) in systems with “weak” interaction is much lower than that with “moderate” or “strong” interaction due to the particle aggregation. The density of neighboring polymer beads increases with the increase of *ε*_np_. Notably, the first peak of rod nanocomposites is higher than that of tetrapod nanocomposites, and this disparity is amplified with the increase of *ε*_np_. This disparity can only be explained that the polymer chain wraps more closely on the tetrapod than rod (effect of steric hindrance) and the smaller surface-area-to-volume ratio on the tetrapod than rod (the aspect ratios of both particles are not two high in current model systems, and thus four rods have a larger surface area than one tetrapod on condition that the rods are not aggregated).

Undoubtedly, for systems with low nanoparticle concentration, *Φ*_n_ (e.g., *Φ*_n_ < 10%), the probability to form percolation network of high-density polymer regions increases with the increase of *Φ*_n_. When polymer beads are not enough to fulfill the vicinity of nanoparticles, the further increase of nanoparticle loading leads to direct contact of nanoparticles and then destroys the percolation network. To find the optimum *Φ*_n_, *Φ*_n_^o^, with the highest probability, the number density of nearest neighbor polymer beads *ρ*^*^_n_ would be an efficient parameter. Figure 9 plots *ρ*^*^_n_ as a function of *Φ*_n_. A higher *ρ*^*^_n_ usually leads to higher percolation probability without considering the effect of fillers. It shows that rod nanocomposites have the highest *ρ*^*^_n_ at *Φ*_n_^o^ around 17.3%, while the tetrapod nanocomposites sample is around 21.4%. This difference of *Φ*_n_^o^ is due to the smaller surface area and larger occupied volume of a tetrapod. Notice that the increase of *ε*_np_ results in the increase of *ρ*^*^_n_ but has no influence on determining the *Φ*_n_^o^. Compared with the probability of percolation as a function of nanoparticle volume fraction in spherical particles nanocomposites [58], the *Φ*_n_^o^ determined by number density of nearest neighbors are all located in the high probability area. As a supplement, a maximum effective *Φ*_n_, *Φ*_n_^o^, exists to form the percolation network, and the percolation probability may decrease by further adding particles beyond *Φ*_n_^o^. In addition, the effect of particle shape should be taken into account to calculate the percolation probability; for example, the tetrapod filler is more efficient to become a “bridge” to connect different stress concentrated areas than spherical filler due to the stronger constraint in the polymer matrix.

## 4. Discussion

3D LSM simulation results show that the tetrapod filler is more efficient to enhance the Young’s modulus of the corresponding polymer composite than rod filler at the same concentration; this disparity is observed in both bare and surface modified particles as fillers and fits well with the experimental data. As for particles with surficial ligands, the mechanical disparity of their nanocomposites can be mainly contributed by the orientation of the strong X-type bonds at the nanoparticle-ligand interface, which is important for amplifying the disparity between rod and tetrapod composites. Here, the discussion related to the disparity of bare particle filled composites is mainly from two aspects: stress transfer and particle agglomeration. The stress transfer in tetrapod filler (Figure 6) could be the key explanation and origin for elastic modulus difference between rod and tetrapod filled composites. The arms in tetrapod are connected via the strong particle bond. Assuming that this strong bond is replaced by a weak spring between polymers, the composites turn to the random rod filled composites and elastic modulus of composites would be dramatically decreased. In addition, the stress on the tetrapod with strong NPI is much higher than that with weak NPI, which leads to a higher elastic modulus. Secondly, as for the particle agglomeration, achieving a uniform dispersion of particle in a polymer matrix is a possible way to get high performance structural materials. Due to the special isotropic shape, the tetrapod will be more loosely packed than a rod in real polymer composites, and thus it is easy to achieve the percolation network of high-density polymer regions. A simple experiment that I performed by LSM is that of the elastic modulus being only slightly increased if the branched tetrapod is aggregated in the center of the cubic box. Particle agglomeration would be an additional effect to lead to the elastic modulus disparity.

## 5. Conclusions

The elastic moduli of polymer composites using rod or tetrapod as fillers are examined by a 3D lattice spring model. Meanwhile, the dispersion and spatial organization in polymer melts of these two fillers are studied by molecular dynamics simulation. The elastic modulus of tetrapod filled composites is much higher than that of rod filled composites. For the bare particles, the disparity is due to the effective stress transfer in tetrapod, and the strong NPI amplifies this disparity. An additional effect is the particle agglomeration in the composites, and molecular dynamics simulation results show that a tetrapod is more likely to achieve uniform dispersion than a rod, and the uniform dispersion would give better mechanical performance.

Based on the comparison of tetrapod and rod filled composites, I conclude that the final mechanical behavior of branched particle filled nanocomposites is controlled by many factors including particle size, shape, intrinsic material properties and NPI. While the dispersion state and efficiency of stress transfer can be modulated by these factors, e.g., on condition that the particle size are comparable with or smaller than the entanglement length of polymer matrix, and the branched shape facilitates the polymer chain wrapping closely on the particle, the stress transfer would become more efficient and the particle probably develops into an entanglement point. In general, this work provides the shape effect on the Young’s modulus of branched filled polymer nanocomposites; it yields guidelines on mechanical property prediction and rational design of polymer nanocomposites.

## Figures and Tables

**Figure 1 polymers-10-01368-f001:**
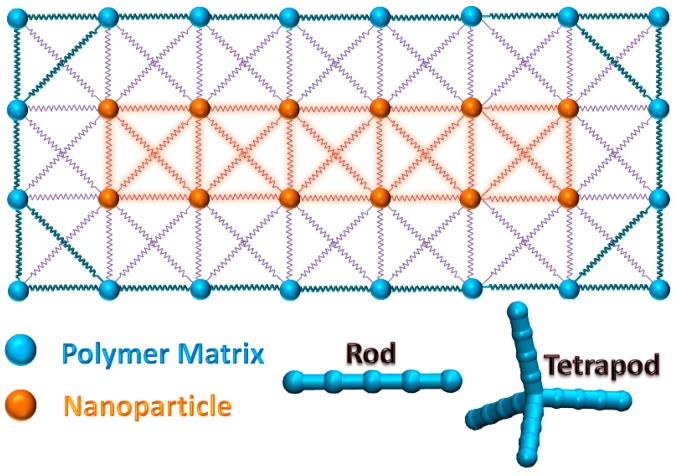
The interconnectivity of the 3D lattice spring model in a typical layer which depicts the branched rod incorporated in a polymer matrix, nearest [100] and next nearest [110] neighbor spring interactions are considered. Three kinds of springs are represented by different colors (different spring constants). The schematic diagrams of rod and tetrapod with aspect ratio of 5 are shown in the right bottom corner.

**Figure 2 polymers-10-01368-f002:**
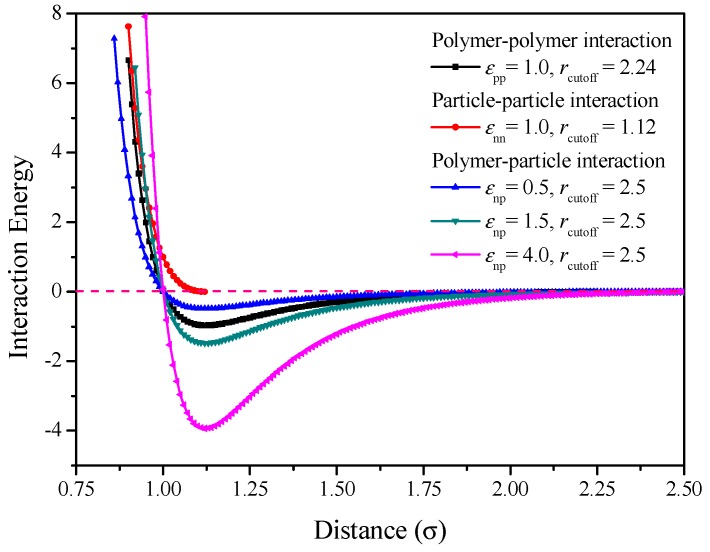
Plot of interaction energy versus distance for various interaction parameters.

**Figure 3 polymers-10-01368-f003:**
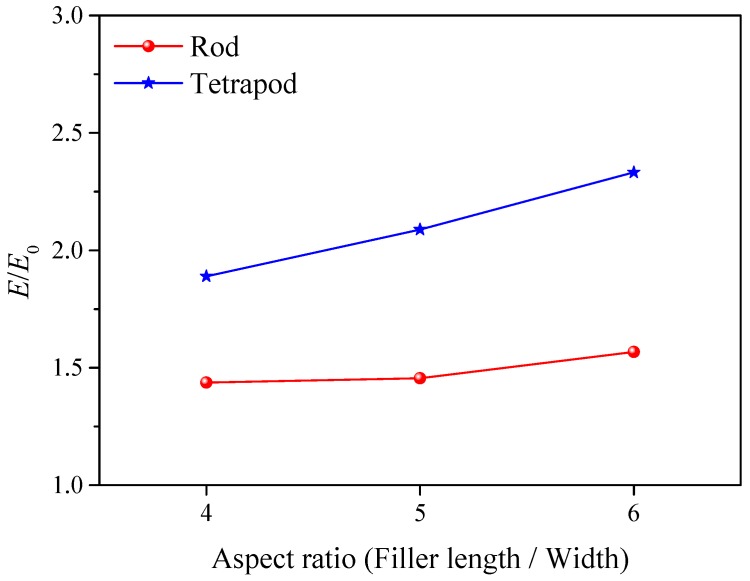
Plot of relative elastic modulus versus aspect ratio for tetrapod and nanorod nanocomposites. Blue lines represent results from the tetrapod filled composites, while red lines represent results from the rod filled composites.

**Figure 4 polymers-10-01368-f004:**
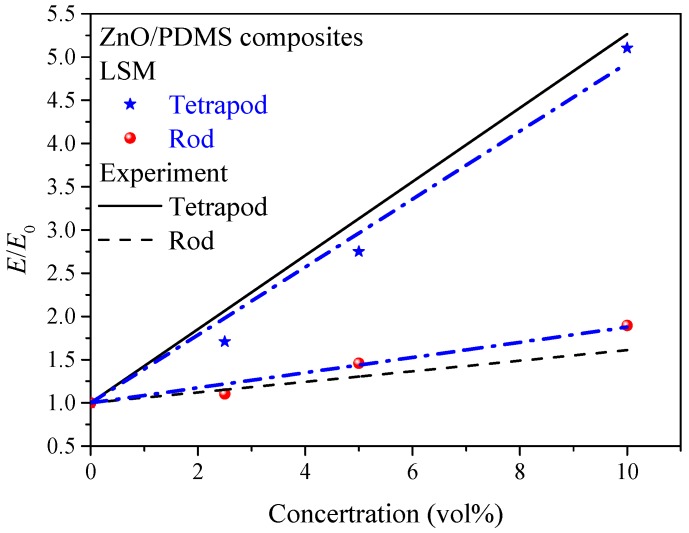
Comparison of experimental results from ZnO/PDMS (Polydimethylsiloxane) composites with simulated data by a lattice spring model. Plot of relative elastic modulus versus nanoparticle concentration for tetrapod and nanorod nanocomposites. Fits are clamped to the (0, 1) point. The experimental data are replotted from Ref. [23] under open access license.

**Figure 5 polymers-10-01368-f005:**
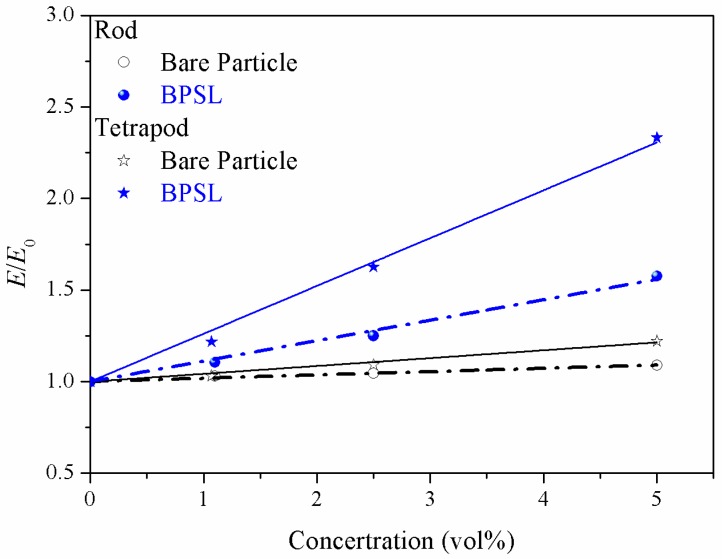
Comparison of bare particles with BPSL (Branched particles with surface ligands) on the relative elastic modulus of composites by lattice spring model. Plot of relative elastic modulus versus nanoparticle concentration. Blue lines represent results from BPSL filled composites, while black lines represent results from bare particles filled composites.

**Figure 6 polymers-10-01368-f006:**
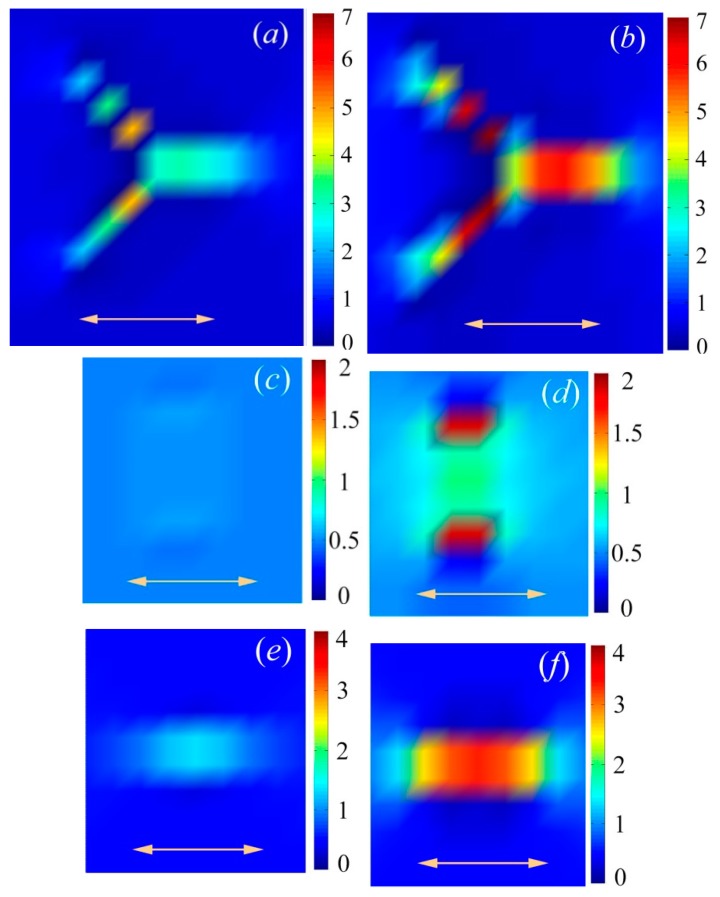
Normal stress profiles through the center of the particles with weak (**a**,**c**,**e**) and strong (**b**,**d**,**f**) NPI. Yellow double arrows indicate the stretching direction. Particles include tetrapods (**a**,**b**), vertically aligned rods (**c**,**d**) and horizontally aligned rods (**e**,**f**). The color bars indicate the value of stress for each bead.

**Figure 7 polymers-10-01368-f007:**
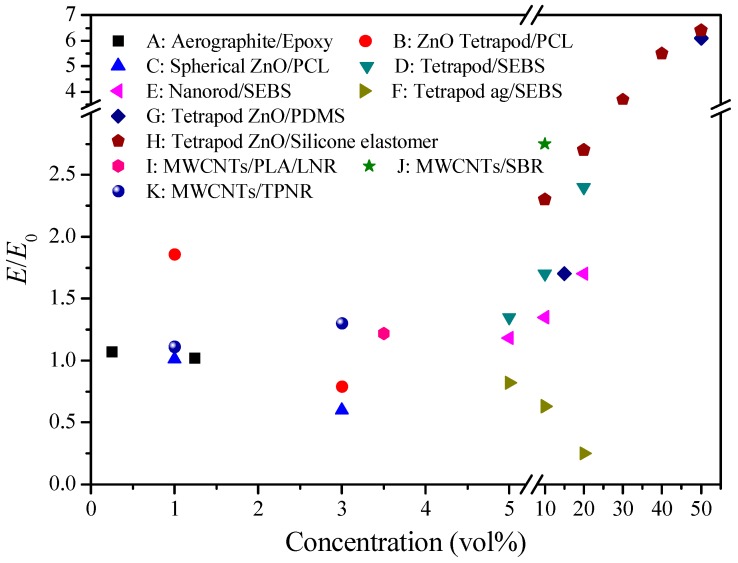
Plot of relative elastic modulus versus nanoparticle concentration for various nanocomposites. (**A**) the aerographite filled epoxy with particle aggregation [49]; (**B**) pristine PCL filled with tetrapod ZnO [50]; (**C**) pristine PCL filled with spherical ZnO [50]; (**D**) SEBS filled with surfaced modified CdSe/CdS tetrapod nanoparticles [44] and simulation results; (**E**) SEBS filled with surfaced modified CdSe/CdS nanorod nanoparticles [44] and simulation results; (**F**) SEBS filled with aggregated CdSe/CdS tetrapod nanoparticles [44]; (**G**) PDMS filled with tetrapod ZnO [51]; (**H**) silicone elastomer filled with tetrapod ZnO [52]; (**I**) PLA/LNR nanocomposite filled with MWCNTs [53]; (**J**) SBR nanocomposite filled with MWCNTs [54]; (**K**) TPNR nanocomposite filled with MWCNTs [55].

**Figure 8 polymers-10-01368-f008:**
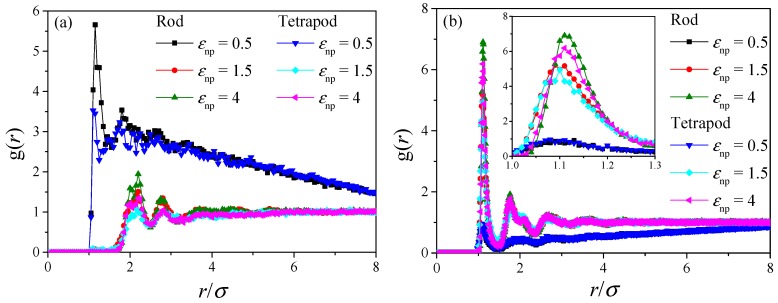
The radial distribution function (RDF) of rod and tetrapod nanocomposites with different polymer-nanoparticle interactions: (**a**) inter-nanoparticle RDF; (**b**) RDF between nanoparticle and polymer matrix; the inset shows the partial enlarged view of the peak.

**Figure 9 polymers-10-01368-f009:**
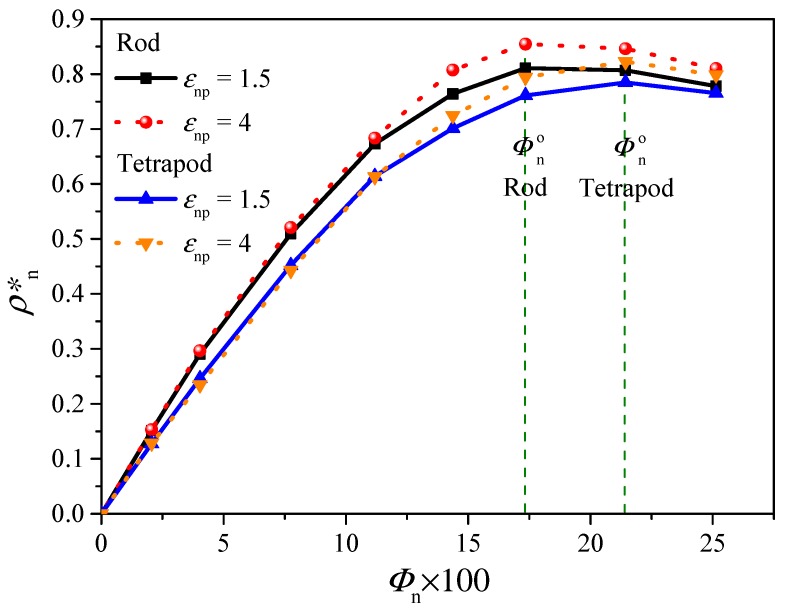
Plot of number density, *ρ*^*^_n_, of nearest neighbor polymer beads versus nanoparticle concentration, *Φ*_n_, for tetrapod and rod nanocomposites.

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
