# Peer review of "Multiscale Simulation of Branched Nanofillers on Young’s Modulus of Polymer Nanocomposites"

_polymers, 2018, doi:10.3390/polym10121368_

Reviewer 1 Report

In this manuscript, Deng compared the topology effect of nanorods on Young’s modulus of polymer nanocomposites. The manuscript is written in a way which can be easily understood. The article can be accepted for publication after these questions are addressed.

1.      In section 3.5, author investigated the balance between epsilonnp and epsilonnn. The interaction between nanoparticle and polymer is nomal. However, the interaction between nanoparticles are pure repulsive. Various cutoffs between different CG beads make me a little confused. Can you add the potentials plots when you compare these interactions.

2.      In line 271, author claimed that 4 rods have larger surface area than 1 tetrapod on condition that rods are not aggregated. However, I think that it depends on the rod length. If the rod is very long, they will have the same surface area.

Minors

Please specify all the C values for each LJ pair

Line 160 what is X type bond?

Line 161 According to the literature value… need reference.

Line 183 need reference.

Line 253 the second one is epsilonnn

Author Response

Thanks for your helpful comments, please see attached file.

Reviewer 2 Report

This manuscript is generally well-written and falls within the scope of Polymers. However, the are some issues that should be addressed before its consideration for publication in Polymers.

-What is the novelty of this work? novelty must be clearly mentioned in the Introduction section. 

-While the manuscript has one author, at some points, "we" were used. 

-What are the advantages and disadvatanges of Lattice spring model?

Author Response

(The authors gave the same response as above.)

Reviewer 3 Report

The work --Multiscale simulation of branched nanofillers on Young’s modulus of polymer nanocomposites-- by Deng and co-workers is very interesting and would be high interest for large number of readers from filler-polymer composite community. The following remarks need to be carefully addressed:

Authors have considered only solid tetrapods as better fillers in contrast to nanorod or bare particles type. However, there are many tetrapod varieties hollow and entangled are available which could be used a filler too, for example, Nature communications 8, 2017, 1215; Scientific Reports 5, 2015, 8839; Advanced Materials 24, 2012, 3486-3490;   Nature Communications 8, 2017, 14982; etc.

Several polymer composites based systems like, ACS Applied Materials & Interfaces 9, 2017, 38000–38007; Nano Letters 17, 2017, 6235–6240; Advanced Materials Interfaces 4, 2017, 1700019; Composites Science and Technology 134, 2016, 226–233; Advanced Materials 25, 2013 1342-1347; have been reported in literature. It would be very helpful to readers if a tabular scheme can presented showing in terms of type of fillers, effect of morphology, solid or hollow (hollow will be better filler as polymer will interlock from inside too), polymer-filler interaction, etc. in the paper. This will directly help readers to speculate the properties of the desired nanocomposite systems. 

At some instances, the paper is hard to follow as model is not very clear, authors are suggested to carefully check the english flow and clarity of the matrix. 

Further references relevant to filler polymer composites and relevant properties and applications could be cited, for example, Nanomaterials 8, 2018, 945 (https://doi.org/10.3390/nano8110945); Materials Today 21 2018, 631-651; and many others.

The manuscript is recommended for publication after a careful revision.

Author Response

Thanks for your helpful comments, please see attached file.

Round  2

Reviewer 1 Report

The author have addressed all the questions.